# Unlocking the Potential of Blockchain Technology in the Textile and Fashion Industry

Sunhilde Cuc 

Department of Textiles, Leather and Industrial Management, University of Oradea, 1 Universitatii Street, 410087 Oradea, Romania; sunhilde_cuc@yahoo.com

**Abstract:** The textile and fashion industry is on the brink of a major disruption, and blockchain technology (BT) presents a promising solution that could transform the industry by facilitating supply chain transparency, traceability, and sustainability. This article explores the potential of BT in the textile and fashion industry, with a focus on its current applications and potential impact. Using case studies and analyzing all announced blockchain projects from January 2017 to January 2023, we examine the diversity of blockchain applications across different aspects of the textile and fashion industry, including smart contracts and payment processing, supply chain tracking, sustainability applications, and customer engagement. The findings suggest an increasing number of companies are adopting BT, and that BT has the potential to revolutionize the T and F industry by creating a more transparent and efficient supply chain, reducing fraud and counterfeiting, and increasing customer confidence in products. We also identified the challenges and difficulties that may arise during the implementation of BT. This article contributes to the literature on BT in the textile and fashion industry, providing critical insights into its potential impact.

**Keywords:** blockchain technology (BT); textile and fashion industry; smart contracts; supply chain traceability; sustainability

## 1. Introduction

The textile and fashion industry has experienced significant growth in recent years, with its global sales reaching a total of 2.14 trillion USD in 2022. Of this amount, 1.53 trillion USD can be attributed to apparel sales [1], while the remaining 610.91 billion USD can be attributed to textile sales worldwide [2]. The term "textile and fashion industry" is employed in the present paper to refer to the broader economic sector encompassing both textile and apparel manufacturing, as well as the fashion design and retail sectors. This terminology recognizes the interconnectedness and interdependence of these various industries and their profound impact on the global economy and society. By utilizing this broader terminology, we seek to more accurately assess the potential applications of blockchain technology within this multifaceted industry. Specifically, we aim to elucidate the ways in which blockchain's distributed ledger system could be leveraged to enhance the efficiency, security, and transparency of various operations within the textile and fashion industry. As the industry continues to expand, it is increasingly important to ensure the sustainability, transparency, and traceability of the supply chain. The rapidly evolving digital economy has placed the traditional textile and fashion industry at the forefront of global dynamic change, creating a landscape characterized by volatility, velocity, variety, complexity, and dynamism [3]. As a result, there is a growing need for digital solutions to address these challenges [4,5]. Blockchain technology has been identified as a potential solution to these challenges because it can provide a secure and immutable record of transactions and data [5–7].

### 1.1. Blockchain Technology (BT)

A lot of research has been conducted in both industry and academia to explore the potential and usefulness of blockchain technology (BT) in various application areas [6–8]. Recently, blockchain technology (BT) has become more widely known and has sparked enthusiasm in many industries, including the textile and fashion industry, due to its potential to improve business processes.

A "blockchain", as the name suggests, is in its simplest terms a "chain" of previously validated "blocks" of transactions, a decentralized database that stores data securely and encrypted in a distributed public ledger. Cryptography is the practice of safeguarding data by encoding it in order to prevent unauthorized access in an environment where security is not assured [6,7]. BT utilizes cryptography to protect the system's integrity by using advanced cryptographic algorithms. These algorithms allow for the secure identification of participants, confidential transactions, and verifiable transaction authenticity. Cryptographic keys are used to sign transactions, ensuring that each participant has validated them and preventing third-party impersonation. Transactions are encrypted and digitally signed so that no part can be altered without being rejected by others in the blockchain. Blocks are added that reference the previous block through a cryptographic signature, forming a chain of blocks [6]. That ensures the trustworthiness of transactions or smart contracts, making them unalterable. It is also known as distributed ledger technology [8–10].

The concept of blockchain was first introduced in 2008, and since then it has been constantly developing [9]. Even though blockchain was born with the advent of Bitcoin, it is no longer limited to cryptocurrencies [10,11]. Today, it has a remarkable effect in many other areas of distributed applications and has achieved significant success in the realms of finance, business, industry, politics, and society. Its features, such as the distribution of data storage among separate nodes and the utilization of consensus algorithms, provide immutability and transparency and eliminate the requirement for a central authority, thus making blockchain technology reliable [8,10,11].

In banking and finance, blockchain offers a secure and efficient way to carry out international payments, enhance capital markets, simplify trade finance, facilitate secure peer-to-peer transactions, and combat money laundering [12–14]. From an industrial standpoint, researchers have observed the possibility of blockchain applications in the supply chain across various industries [15], including textiles and fashion [16,17].

The integration of digital currencies, smart contracts, and distributed data storage through blockchain technology is enabling the emergence of novel decentralized structures, such as decentralized autonomous organizations, which are regulated by source codes to determine their governance structure [18].

Recent research [19] has revealed that 44 of the top 100 public companies by market capitalization across six major sectors are actively utilizing BT. The technology, media, and telecom sector is leading the way, with 36% of these companies belonging to this sector, including Meta (NASDAQ: META), Salesforce (NYSE: CRM), Adobe (NASDAQ: ADBE), Verizon (NYSE: VZ), and Nvidia (NASDAQ: NVDA). The consumer and retail (20%) and basic materials and industrials (20%) sectors are also represented, with participants such as UPS (NYSE: UPS), PayPal (NASDAQ: PYPL), Visa (NYSE: V), Walmart (NYSE: WMT), McDonald's (NYSE: MCD), and Nike (NYSE: NKE). Despite occasional setbacks caused by regulatory and macroeconomic hurdles, the blockchain and cryptocurrency industry is continuing to grow in adoption and use cases among large global institutional players.

Of the 100 companies in the global top 100, 63% are from the United States, while 12% are from China (including the Hong Kong Special Administrative Region). Not all of these companies are actively utilizing blockchain technology; however, 86 of them are actively pursuing blockchain-related solutions for their business needs [19] (Figure 1).

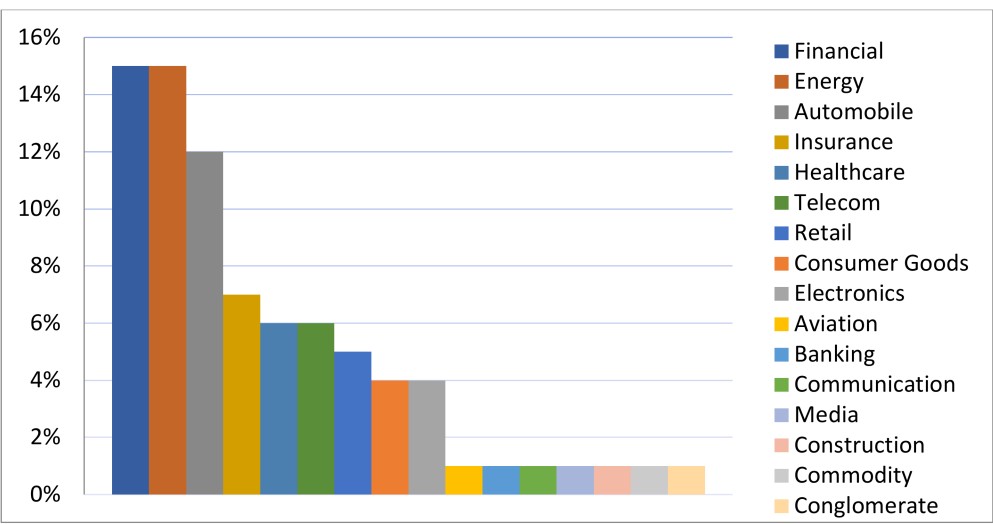

**Figure 1.** Blockchain adoption among Fortune 100 in 2022.

Contrary to popular belief, blockchain technology (BT) is not in the public domain like the Internet; instead, it is patentable, and many of the newer blockchain platforms have been patented. In 2021, 25.2 billion USD of venture capital was allocated to blockchain startups worldwide, representing a 713% increase from 3.1 billion USD in 2020. Global venture funding for blockchain and crypto companies achieved a new peak of 26.8 billion USD in 2022, mainly due to a strong first half. However, as the year progressed, the crypto winter combined with macroeconomic pressures caused three consecutive quarters of decreases in funding and transactions [20].

There has been considerable focus on the swift advancement of blockchain technology (BT) in the application, particularly among academics. Scholars are dedicating a great deal of study to the topic of blockchain. Despite the rapidly increasing popularity and interest in this technology, there is limited knowledge regarding the current application and utilization of blockchain in the textile and fashion industry.

### 1.2. Textile and Fashion Industry

Fashion is one of the most influential industries in the world, playing a major role in the global economy. It is a major contributor to the world economy, and if it were to be ranked alongside individual countries' GDP, the global fashion industry would be the seventh-largest economy in the world [21]. The textile and fashion industry is characterized by geographically dispersed production and rapid market-driven changes, providing employment opportunities to millions of workers worldwide, particularly for women. Around the world, one out of every six workers is employed in the apparel industry, and women make up 80% of the workforce in the supply chain. Due to the scale and profile of workers employed, the sector has the potential to make a significant contribution to economic and social development. Approximately 3000 billion textile and garment companies are entering the market daily [22].

Textiles and clothing is a diverse sector that plays an integral role in the global manufacturing industry, with a value of more than 2.14 trillion USD and employing over 75 million people worldwide [1,2]. In Europe alone, it employs 1.7 million people and generates a turnover of EUR 166 billion [23]. The sector has seen remarkable growth over the past decades, and the forecast is also optimistic. The global apparel market experienced a gradual increase in revenue from 2015 to 2020, when the coronavirus (COVID-19) pandemic had a significant impact on retail. In 2022, the revenue of this market was estimated to be 1.53 trillion USD, and it is projected to reach nearly 2 trillion USD by 2027 [24]. The global textile market has exhibited a growth from USD 573.22 billion in 2022 to USD 610.91 billion in 2023, corresponding to a compound annual growth rate (CAGR) of 6.6%. This growth

has been interrupted by the Russia–Ukraine war, which has significantly impeded global economic recovery from the COVID-19 pandemic, particularly in the short term. The hostilities have resulted in economic sanctions being imposed on various countries, as well as a sharp increase in commodity prices and supply chain disruptions, leading to inflation across goods and services and impacting many markets worldwide. Despite these challenges, the textile market is expected to expand to USD 755.38 billion by 2027, reflecting a CAGR of 5.5% [2]. The revenue data of the textile and apparel market worldwide from 2015 to 2027, presented in Figure 2, have been collected from sources [2,24], as indicated in their own presentation.

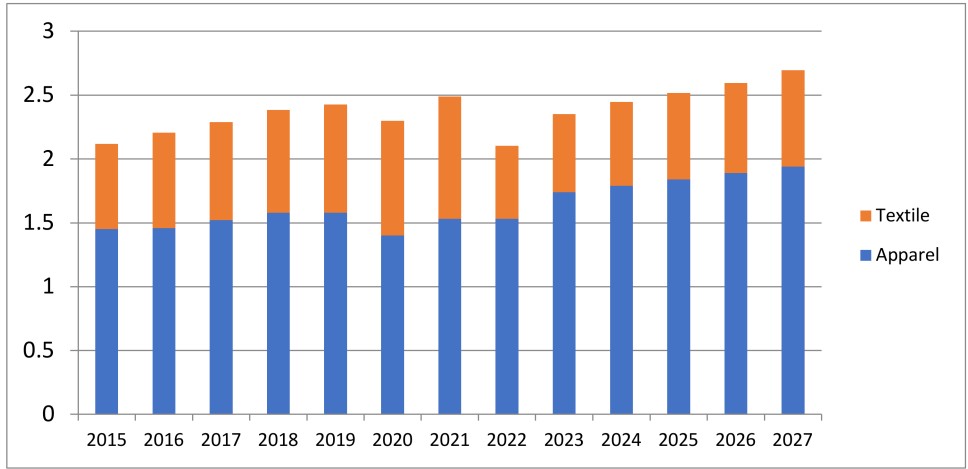

**Figure 2.** Revenue of the textile and apparel market worldwide from 2015 to 2027 [2,24].

The configuration of the textile and fashion supply network varies in terms of complexity, geographical spread, and size. A wide range of natural (e.g., cotton, wool, silk) and synthetic fibers (e.g., nylon, polyester, rayon, etc.) are used in garment production. Despite the geographical dispersion of the industry and the diversity of materials used, five main value-adding stages are evident in most textile and apparel supply chains: (1) raw material sources (e.g., farm, forest, fiber plant, etc.); (2) textile companies (e.g., fiber producers; spinners; fabric makers); (3) garment manufacturers (e.g., designing, cutting, sewing, ironing, etc.); (4) export network (e.g., intermediaries as trading entities and logistics providers involved in buying, selling, and transporting textile and apparel products across the principal value-adding activities and/or carrying out specific processing activities); and (5) brands and retailers (see Figure 3) (adapted from [25]).

Depending on the specific product being offered, the production chain can differ significantly, emphasizing one aspect over the other. For example, the luxury fashion industry is distinct from the mass-produced apparel retail industry. Luxury ensembles and accessories are often exclusively designed and manufactured for special orders, such as couture houses, and luxury brands often control the design and quality of their products by owning the entire supply chain or opting for the most reliable manufacturer [26]. The mass-produced apparel and retail industry, on the other hand, comprises the most affordable-fashion brands and labels, which operate on a different supply chain model compared with luxury brands [27]. Fashion brands that target mass-produced apparel aim to manufacture products at the lowest cost to increase their profit margin. Thus, the potential application of BT in the luxury segment can be used to create a crypto-legal structure of endogenous protective laws administered through a decentralized system of self-executing smart contracts which can fill the gap in the existing intellectual property regime. This can deter counterfeiting, while the mass production sector, so-called fast fashion, can enhance efficiencies and ensure cost savings through the use of smart contracts and secure payment.

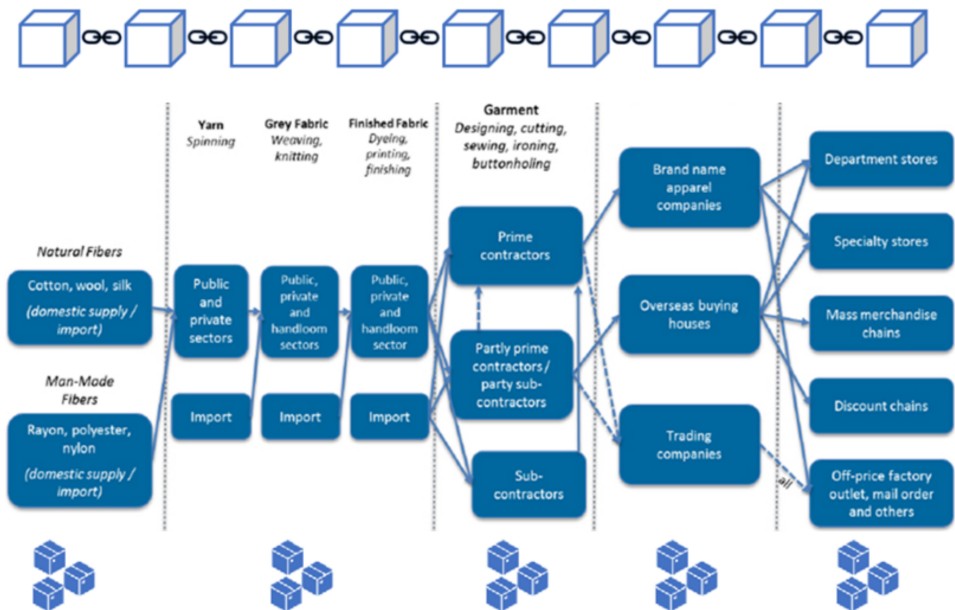

**Figure 3.** Integration of blockchain technology in the textile and fashion supply chain.

Fashion companies that can adapt to increasing complexity by revising their operating models and altering their strategies for supply chains, sales channels, and digital marketing will be best positioned to withstand such a competitive market [21].

Even though BT has been implemented in several industries [28,29], the literature on the subject indicates that no sufficient attention was paid to adopting blockchain in the textile and fashion industry. In the last period, blockchain has been increasingly considered for various applications in the textile and fashion industry, ranging from smart contracts, supply chain tracking operations [30], and supply chain management [31], to product authentication and payment processing. However, evidence from practice is still scarce regarding why, where, and how organizations seek to apply this technology in their supply chain [16].

This study lays the foundation for further theoretical perspectives and empirical research to investigate the characteristics of the textile and fashion industry and the importance of various types of blockchain applications in the supply chain.

## 2. Materials and Methods

The application of blockchain technology (BT) in different industries continues to develop, with an increasing number of blockchain projects being announced. To achieve our research aim, we focused on analyzing announced blockchain projects across the textile and fashion industries from January 2017 to January 2023. To capture the diversity of blockchain applications across different contexts of this complex sector, we examined the following aspects: smart contracts, payment processing, and supply chain tracking including product authentication, sustainability application, and customer engagement.

The selection of the textile and apparel industry for this research was based on several factors. Firstly, the sector is rapidly advancing in its experimentation with blockchain technology [32]. Secondly, the supply chain configuration, product, and business process characteristics vary significantly across this industry, as do the regulatory regimes and market requirements under which they operate [33,34]. Thirdly, it is one of the most challenging sectors for sustainable development, with numerous social and environmental issues [35,36]. Furthermore, the industry is highly fragmented and globalized, and production, shipping, and logistics is a major sector responsible for managing material flows across complex and global supply chains and is subject to various regulations, business processes, and management requirements.

This paper presents two distinct applications of blockchain technology (BT) in the textile and fashion industry. In the first part, we will discuss smart contracts, the connection between blockchain and them, and the potential impact of smart contracts on the textile and fashion industry. The second part will focus on the use of blockchain in the supply chain in the textile and fashion industry. It will emphasize BT's impact on the complex production chain, and two aspects of industry interest will be highlighted: product authenticity tracking and blockchain benefits and barriers in the sustainable supply chain. In the third section, we review the reported blockchain applications in the textile and apparel industries and provide a synthesis of the main projects adopted by companies in the textile and fashion industry. Finally, we briefly conclude the study.

This research will employ data collection and data analysis methods to further explore these issues. A comprehensive review of data from media sources and the practical literature was conducted to gain an understanding of the use of BT within the textile and fashion industry. The sources for this research were accessed through multiple databases, including Google Scholar, ScienceDirect, IEEE, Emerald, and Scopus, to minimize any potential bias in selection [37].

## 3. Results

### 3.1. Smart Contracts

With the rapid advancement of artificial intelligence, which has significantly advanced and improved computational law, smart contracts have once again become a focus of attention. Faced with massive changes in consumer behavior and disrupted supply chains due to the COVID-19 pandemic, the textile and fashion industry quickly adapted by accelerating digitization and facilitating the adaptation to online sales strategies, the implementation of smart contracts, and the establishment of secure payment processes [38]. Smart contracts have become popular recently in diverse industries, such as insurance, energy, real estate, financial services, health care, entertainment, etc. [39,40].

Although blockchain technology is being explored in many areas, the utilization of smart contracts is one of the most significant features of blockchain applications [40,41], allowing for the execution of trusted transactions without the need for third-party intermediaries.

Smart contracts are not exclusively linked to blockchain technology; they are often referred to using a variety of terms, such as "Digital Contract", "Smart Legal Contract", or "Smart Contract Code". Generally, all definitions of smart contracts involve some form of automated, self-executing transaction [40–42]. A smart contract is a piece of code stored on the blockchain that runs automatically when certain conditions and rules are met. It encodes a business agreement between two parties, which is then verified and signed by them before being uploaded to the network and does not require a lawyer to be involved.

These contracts are created to guarantee that the terms of the agreement will be followed without having to go to court. When the terms and conditions are met, the smart contract will automatically execute and carry out the tasks it was programmed to do, such as releasing funds, making payments, and transferring assets. Automation takes away the need for humans to make decisions when it comes to carrying out the contract, regardless of the outcome. This is achieved through the implementation of cryptographic hashes, digital signatures, consensus algorithms, timestamps, and incentive policies, thus enabling peer-to-peer transactions in a distributed environment without the need for mutual trust.

A variety of interpretations of smart contracts have been suggested. One of the first definitions of a smart contract presents it as "a set of promises, specified in digital form, including protocols by which the parties fulfill these promises" [42]. Generally speaking, smart contracts can be defined as computer programs that enforce the terms and conditions of a particular agreement or contract between two or more parties on a blockchain using software codes and computational infrastructure. By definition, a contract is a legally enforceable agreement, and in the case of smart contracts, the agreement will be enforced not by public law enforcement, but by the terms and mechanisms established within the contract itself. A smart contract does not need to be enforced by a government, but it can

help prevent misunderstandings between people. It is a way to ensure that all parties involved in a deal fulfill their obligations [43]. Scholars classify smart contracts into two categories based on their legality: strong and weak [44]. Weak smart contracts can be altered without any additional cost, while strong smart contracts cannot be modified or the cost of modification is too high to be practical. Traditional law enforcement cannot intervene in the execution of strong smart contracts, regardless of whether it is performed by a third party or a judge.

Ref. [45] proposed a taxonomy of smart contracts into five categories, which describe their intended application domain.

- Financial contracts that manage, gather, or distribute money as a preeminent feature; certify the ownership of a real-world asset, endorse its value, and keep track of trades or implement crowdfunding services; gather money from investors in order to fund projects; provide an insurance on setbacks that are digitally provable.
- Notary contracts that exploit the immutability of the blockchain to store some data persistently, and in some cases to certify their ownership and provenance, or allow users to write the hash of a document on the blockchain so that they can prove document existence and integrity; associate users with addresses in order to certify their identity.
- Game contracts which implement games of chance or skills (e.g., Lottery).
- Wallet contracts that handle keys, send transactions, manage money, and deploy and watch contracts in order to simplify the interaction with the blockchain.
- Library contracts that implement general-purpose operations to be used by other contracts.

There are a growing number of smart contract platforms available, each with its own unique features that make it suitable for certain applications. The top smart contract platforms in 2023 include Ethereum (or Ether or ERC-20), widely considered to be the best general-use smart contract platform that can be used for everything from ICOs to facilitating smart contract use with almost any kind of decentralized application; Hyperledger Fabric, established by the Linux Foundation in December 2015; NEM, launched on 31 March 2015 [46]; Stellar, one of the oldest smart contract platforms founded in 2014; Waves, launched in June 2016, an opensource platform designed to facilitate token operations; Solana, developed in 2017 to address issues faced by the Ethereum platform; Avalanche; Polkadot; Algorand, and more.

Some fashion brands that use the Ethereum platform for smart contracts include Nike, Adidas, Puma, Burberry, Gucci, Prada, Louis Vuitton, and Versace.

### 3.2. Traceability and Tracking of Textile and Fashion Supply Chain

Blockchain technologies can be used for traceability and tracking in the supply chain in a variety of areas. Traceability involves tracing the origin of a product or material, while tracking involves tracking the current location and status of a product or material. Traceability can be used to ensure that the product or material is from a known, trusted source and that it has been handled correctly, while tracking can be used to monitor the product or material in real time and ensure that it is delivered to the correct destination in a timely manner.

The academic literature concerning the utilization of blockchain in the supply chain began to appear only in 2016 [47,48]. Among the initial studies, scholars explored the potential of blockchain in service systems by enabling the sharing of information and the collaborative production of value in a reliable and transparent environment [49].

The traceability of the textile and fashion industry is becoming increasingly important as consumers demand greater transparency regarding the origin, history, components, and perceived quality of the products they purchase. Additionally, traceability can help to reduce the environmental impact of the industry by providing information on the sustainability of the production process and ensuring that workers in the supply chain are treated fairly and their rights are respected. From an industry perspective, traceability can help to improve the efficiency of the production process and reduce costs, as well as

identify potential problems and take corrective action to prevent them from occurring again. Furthermore, traceability can help to ensure that the products meet the required standards and that the quality is consistent. Finally, traceability can help to protect the industry from counterfeiting and other fraudulent activities.

Blockchain technology can be used to track the entire supply chain from raw materials to finished products [16,50,51]. As presented in the introduction, the textile and fashion industry is mainly composed of heterogeneous small- and medium-sized companies, which are often highly specialized in certain processes, forming global supply chains for the entire range of activities, from sourcing raw materials to delivering finished products to customers. Among the factors that tend to increase its complexity, the large number of raw materials used in the preparation of fibers and the diversity of manufacturing steps necessary to obtain them can be highlighted. All these factors make traceability almost impossible [52].

Figure 3 presents that in the T&C supply chain, the upstream partners take the raw materials in various forms as inputs from the suppliers. They perform various operations to create the final product, which is then passed on to the next supply chain partner. This process is repeated by various supply chain partners until the product is supplied to the retailer. Consequently, much information is generated at each stage of the supply chain that needs to be collected and managed properly. This information is a crucial part of the supply chain, and each partner must work to control its flow and protect confidential information. All the information can be recorded, but only essential information should be shared on the distributed ledger. Traceability and tracking are key elements of supply chain management in the textile industry, with traceability referring to the ability to track the movement of a product or material throughout the supply chain, and tracking referring to the monitoring of specific attributes or characteristics of a product or material. The importance of these factors lies in their ability to ensure transparency and accountability in the supply chain, enabling companies to identify and address issues such as unethical labor practices, environmental violations, and product quality concerns. Table 1 presents a list of the factors related to tracking and traceability in the textile and fashion supply chain as identified in the literature [16,47,48,51,52].

**Table 1.** Tracking and traceability factors.

| Tracking Factors | Traceability Factors |
| --- | --- |
| Location of material sources | Origin of raw materials |
| Processes undergone by the raw material | Source of products |
| Quality of raw material | Compliance with standards |
| Location of goods | Manufacturing location |
| Production and delivery dates | Certification of quality |
| | Quantity of finished products |
| Payment tracking | Fraud prevention |
| Product authenticity | Counterfeiting prevention |
| Timelines for delivery | Processing data |
| Cost of production | Environmental impact of production processes |
| Shipping status | Circular economy issues |

The utilization of blockchain technology in supply chain management is widely recognized, as it securely records and stores all transaction data of stakeholders in the supply chain. Due to the intricate nature of supply chains often being used to conceal the source, tracking, and legitimacy of fashion and textiles, the use of BT makes supply chain information more transparent and the distribution of information more equitable. Blockchain has the potential to significantly influence supply chain management, its related processes, and the governing structures associated with them [53]. Long and intricate supply chains can be tracked with relative ease and efficiency by recording essential data in the blockchain throughout the product's journey from the source of raw materials to the manufacturer to the customer. Using BT to interconnect distributed ledgers, databases, and stakeholders in the supply chain can improve effectiveness and guarantee cost and

time savings, and also allows manufacturers to monitor the quality of their products and ensure that they are meeting customer expectations. The literature has yielded essential traceability information for the textile and clothing supply chain, which should be stored and accessible to any stakeholder. This information has been divided into four categories: product, quality, process, and social–environmental [52].

### 3.2.1. Brand Authentication

The utilization of blockchain to deter counterfeiting has been acknowledged in certain industries, such as the food [54], automotive [55,56] pharmaceutical [57,58], and fashion industries [59–61]. One of the major challenges faced by the fashion industry is the influx of counterfeit products in the marketplace. The issue of counterfeit goods has become a major concern for governments, economists, and business leaders. According to the Organization for Economic Cooperation and Development (OECD), the value of counterfeit and pirated products worldwide was estimated to be around USD 1 trillion in 2013 and is projected to reach nearly 3 Trillion USD by 2022 [62]. The value of counterfeit and pirated goods seized by customs globally was estimated to be USD 509 billion, up from USD 461 billion in 2013, representing 2.5% of world trade. In 2019, imports of counterfeit and pirated products into the EU amounted to as much as EUR 119 billion (USD 134 billion), which represents up to 5.8% of EU countries' imports [63]. The European Union experienced an increase from 5% to 6.8% of imports from non-EU countries. As these figures do not incorporate domestically produced and circulated knock-off goods or pirated digital products distributed online, the actual amount of counterfeiting and piracy is thought to be much higher. Mainland China and Hong Kong were the primary producers of these fake goods, with the United Arab Emirates, Turkey, Singapore, Thailand, and India also contributing substantially. The United States suffered the most economic losses due to counterfeiting, followed by France, Italy, Switzerland, and Germany [64].

Blockchain technology is used to authenticate textile products by creating a digital record of the product's origin, production process, and distribution. This record is used to track the product throughout its lifecycle, ensuring that it is genuine and has not been tampered with. Whenever members of the supply chain upload data concerning a product, it is stored in the blockchain network in a perpetual state. Distinctly different from traditional software-based tracking systems, the blockchain network is not managed by a single entity; instead, the information is located in a decentralized database that is shared among numerous nodes. All of the information is encrypted separately, meaning that any alterations to the data must be accepted by the remaining nodes in the network—an endeavor that a perpetrator of fraud would find difficult, if not impossible, to accomplish. On a practical level, blockchain technology simplifies the process of identifying cases of fraud. With a permanent record of textile and fashion products that traverse the supply chain, companies are able to compare real-world items to their digital counterparts. Any discrepancies between the two will immediately alert companies to possible issues. Additionally, blockchain technology can be used to store product information such as size, color, and fabric type, allowing customers to easily verify the authenticity of the product. This helps to reduce counterfeit products and protect the brand's reputation. In order to drive innovation in the blockchain space and create a platform for collaboration and knowledge sharing, and to develop the applications of blockchain technology and raise the standards of luxury, Aura Blockchain Consortiu was created in April 2021 [64]. This is a collective of leading companies, (e.g., LVMH, Prada Group, and Cartier) universities, and research institutions that are working together to develop and deploy blockchain-based solutions. The consortium is focused on developing and deploying blockchain technology to create a secure and trusted digital infrastructure for businesses and governments [65]. LVMH, the French luxury goods conglomerate, who owns over 70 luxury fashion brands, including names such as Dior, Fendi, Givenchy, Kenzo, and Celine, has implemented a blockchain-based solution in partnership with Microsoft and ConsenSys to verify the authenticity of its

luxury products. The solution enables customers to scan a QR code on the product using their smartphone, which then verifies the product's authenticity on the blockchain.

The advantages of traceability extend beyond the company–consumer relationship, creating trust and transparency between members of the supply chain, including companies and their suppliers and distributors.

### 3.2.2. Blockchain and Sustainability Issues

Scholars have conducted exploratory studies into the potential of blockchain technology to facilitate sustainability initiatives [66]. Researchers have assessed the advantages and drawbacks of integrating blockchain into sustainable supply chains [59]. From a sustainability and corporate image standpoint, it is becoming increasingly necessary to trace the effects of production on society, the environment, and the economy, which has made blockchain a focal point in providing a deeper level of understanding and assurance of operations. The textile and fashion industry has become a focus of media scrutiny regarding sustainability and circular economy issues; however, obtaining accurate data remains difficult due to the globalization of fashion supply chains and the historically limited attention paid to lifecycle sustainability issues in comparison with other industries [67,68]. In the past, many companies have experienced damage to their corporate image due to the revelation of a corrupt supply chain that included forced labor, armed conflict, or toxic emissions. Blockchain could be a barrier to such activities, as the technology would need to include data points that are specifically designed to identify these malicious practices. In 2022, the Swedish company TrusTrace launched its new blockchain-based solution for supply chain traceability, TrusTrace Certified Material Compliance, in an effort to combat misinformation and improve transparency and traceability within the fashion industry. This solution is an extension of the existing product traceability and supply chain transparency platform, and it is intended to be a comprehensive one-stop shop for material compliance. Early input into developing the platform has been provided by major sports brands such as Adidas, Decathlon, and Filippa K, in line with their ambitious sustainability goals for the coming years [69].

### 3.2.3. Customer Engagement

Blockchain technology can also be used to enhance customer engagement in the textile and fashion industry. Customers are increasingly demanding transparency and sustainability from brands [31], and blockchain technology can provide a platform for delivering this information. For example, a blockchain-enabled platform could enable customers to track the environmental impact of a product throughout its lifecycle, from raw material extraction to disposal. This would enable customers to make more informed purchasing decisions and incentivize companies to improve their sustainability practices. For example, The Fabricant, a Netherlands-based digital fashion house, has launched a blockchain-enabled platform called "Immaterial", which allows customers to purchase and own digital fashion items as non-fungible tokens (NFTs). For NFTs, it partnered with Vogue and Diesel. The platform uses BT to ensure its customers of the ownership and authenticity of digital fashion items [70].

The information presented in the article is summarized in Table 2, which outlines the successful adoption of blockchain technology (BT) by various organizations and companies. The table provides examples of the areas of application and the corresponding solutions adopted by these entities.

**Table 2.** Examples of Organizations and Companies Successfully Utilizing Blockchain Technology: Areas of Application and Solutions Adopted (2017–2023).

| Area of Application | Example Company or Association | Solution | Source |
|---|---|---|---|
| Smart contracts | LVMH | +Accept cryptocurrency as payment | [71] |
| | Gucci | | [72] |
| | Philipp Plein | | [72] |
| | Adidas | | [73] |
| | Inditex (Zara) | | [74] |
| Supply Chain Management | UK Fashion and Textile Association (UKFT) | Enabling product traceability and improving transparency in the supply chain | [75,76] |
| | AURA Consortium | Is supporting the first global blockchain solution dedicated to the luxury goods industry, promoting the use of a single global blockchain platform open to all luxury brands to provide consumers with additional transparency and traceability | [64,65] |
| | Arianee Consortium | Is a decentralized, opensource protocol that leverages blockchain technology to create unique digital identities for luxury goods, including fashion items. The Arianee protocol enables brands and retailers to track the ownership, provenance, and authenticity of high-end fashion products throughout their lifecycle. | [77] |
| | The Woolmark Company | Tracking the origin of wool fibers from farm to fashion to ensure that the wool fibers are ethically sourced and traceable. | [78] |
| | Lenzing Group | Tracing products across the supply chain (manmade cellulose fibers) | [79,80] |
| | Chargeurs Luxury Materials | Ensuring product quality, sustainability and traceability across the supply chain (wool) | [81] |
| | Gucci | Tracing products across the supply chain | [82] |
| | LVMH | Tracing products across the supply chain | [75,76] |
| | Stella McCartney | Collaboration with Bolt Threads and Evrnu to create a "regenerated" cashmere sweater made from recycled materials, tracked using a blockchain platform | [83] |
| | H&M | Implementation of a blockchain-based system for tracking and sharing information about suppliers and their sustainability practices | [84] |
| | C&A | Tracing organic cotton from the farm to the ginning process, with a plan to extend it to the consumers | [85] |
| | Inditex (Zara) | Data securitization and tracing products across the supply chain | [86] |
| | Levi Strauss & Co. | Levi Strauss & Co. started testing the blockchain version of the Worker Well-being survey with SHINE to better understand if, in fact, "what's good for workers is also good for business". | [87] |
| | Adidas | Supply chain traceability for sustainable materials | [69,88] |
| | Decathlon | Supply chain traceability for sustainable materials | [87,88] |
| | Nike Inc. | Supply Chain Data Collection Tracking and verifying the movement of cotton fiber across the supply chain | [89,90] |
| Brand Authentication | LVMH Prada | Blockchain-based solution in partnership with Microsoft and ConsenSys for verifying the authenticity of luxury products (AURA platform) | [64,91] |
| | HUGO BOSS | In collaboration with ASTRATUM create Tracey, a blockchain-based system to monitor items in their supply chain and validate their genuineness | [91] |

**Table 2.** *Cont.*

| Area of Application | Example Company or Association | Solution | Source |
|---|---|---|---|
| Customer Engagement | The Fabricant | Is a Netherlands-based digital fashion house that has launched a blockchain-enabled platform called "Immaterial", which allows customers to purchase and own digital fashion items as non-fungible tokens (NFTs) For NFTs, it partnered with Vogue and Diesel The platform uses BT to ensure its customers that the ownership and authenticity of digital fashion items are transparent and immutable | [70] |
| | Levi Strauss & Co. | Pilot project using blockchain technology to enable customers to scan a QR code on their jeans and access information about the production process, materials used, and sustainability practices | [87] |
| | Nike Inc. | Launched a digital community and experience hub and a home for virtual creations and products | [92,93] |
| | Christian Dior | Uses blockchain for their loyalty programs | [94] |

## 4. Conclusions

After examining the potential of BT in the textile and fashion business ecosystems, several conclusions can be drawn:

- Transformational Potential: BT has the potential to revolutionize the textile and fashion industry by creating a more transparent and efficient supply chain. This transformation can improve traceability, reduce fraud and counterfeiting, and instill greater customer confidence in purchased products.
- Infrastructure for Enhanced Connectivity: BT provides a robust infrastructure that facilitates the connection of intricate networks and databases in the industry. This enables simultaneous and irreversible updates across all interconnected databases, streamlining processes and allowing for automation where necessary.
- Collaborative Adoption for Success: Successful implementation of BT in the textile and fashion industry relies on collaboration between different stakeholders. Manufacturers, suppliers, retailers, and consumers must work together to establish a unified blockchain network that effectively tracks and verifies product authenticity.
- Efficiency and Cost Savings: The adoption of BT offers the potential for increased efficiency and cost savings in supply chain management. Through streamlined processes, reduced paperwork, and the elimination of intermediaries, operational efficiency is enhanced, leading to tangible cost benefits.
- Sustainability and Transparency: BT contributes to improved sustainability by fostering transparency in supply chain practices. The ability to verify ethical and sustainable practices empowers consumers to make informed choices, promotes responsible production and consumption, and supports sustainability initiatives in the industry.

While significant benefits can be achieved through BT implementation, it is important to acknowledge and address the challenges that accompany it. Technical limitations, regulatory issues, and concerns over data privacy and security need to be carefully considered and overcome [95]. The successful adoption of BT in the textile and fashion industry necessitates thoughtful examination of these challenges, as well as close collaboration among various actors, including designers, manufacturers, retailers, and policymakers. By working together, these stakeholders can overcome obstacles and fully realize the potential benefits of BT adoption.

Further research is required to fully comprehend the extent to which blockchain contributes to sustainability initiatives. Additionally, potential risks associated with implementing BT should be taken into account to ensure that its utilization is beneficial for the environment.

In conclusion, the adoption of BT in the textile and fashion industry offers substantial benefits for all stakeholders involved. By leveraging its potential, such as increased efficiency, cost savings, improved sustainability, and enhanced supply chain transparency, the industry can thrive. However, it is imperative to address the challenges at hand, including data privacy concerns and potential disruptions to the workforce, through close collaboration among stakeholders. By doing so, the textile and fashion industry can fully harness the transformative power of BT.

**Funding:** This research received no external funding.

**Institutional Review Board Statement:** Not applicable.

**Informed Consent Statement:** Not applicable.

**Data Availability Statement:** Available on request.

**Conflicts of Interest:** The author declares no conflict of interest.

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
