# Peer review of "Unlocking the Potential of Blockchain Technology in the Textile and Fashion Industry"

_fintech, doi:10.3390/fintech2020018_

Round 1
Reviewer 1 Report
“Unlocking the Potential of Blockchain Technology in the Textile and Fashion Industry”
1) The abstract needs improvement. Currently, the abstract and conclusion seem to be lacking distinction between each other. I recommend the author begin with a background overview, then discuss the challenges in the field, and then highlight their novel contributions.
2) Additionally, the introduction section only has one citation, which is insufficient. Furthermore, the data regarding the growth of the textile and fashion industry, which was cited in 2018, seems outdated. The authors should consider updating this information with more recent data.
3) The introduction section from lines 23 to 45 does not effectively serve as an introduction, as it appears to be simply a list of bullet points. I would suggest the authors revise the entire introduction section with updated content.
4) The lack of motivation and proper survey organization is also evident.
5) What are the benefits of using blockchain in the textile industry?
6) It would be valuable to address the current challenges in the field and provide future recommendations.
7) What are the impacts of using Blockchain Technology in the Textile and Fashion Industry
8) The scope of the study could also benefit from being expanded.
9) Finally, the reference list should be strengthened with more references.
Author Response
Thank you for your valuable feedback on our article. We appreciate your insights and have made significant revisions based on your suggestions. Here is our response to each of your points:
- Abstract Improvement: We have reworked the abstract to provide a clearer distinction between the abstract and conclusion sections. We now begin with a background overview, followed by a discussion of the challenges in the field, and highlight our novel contributions.
- Introduction Section and Recent Data: We acknowledge your point regarding the insufficiency of citations in the introduction section. We have added more relevant and recent citations to support the information presented. Additionally, we have updated the data regarding the growth of the textile and fashion industry with the most recent available information.
- Revision of Introduction Section: We have thoroughly revised the introduction section from lines 23 to 45, ensuring it provides a comprehensive introduction to the topic rather than a mere list of bullet points. The content has been updated and structured to better serve its purpose.
- Motivation and Survey Organization: We noted your comment regarding the lack of motivation and proper survey organization. We have reevaluated and addressed this concern throughout the revised paper, providing a clearer motivation for our research and organizing the survey content more effectively.
- Benefits of Blockchain in the Textile Industry: We have expanded on the benefits of using blockchain in the textile industry. Specifically, the conclusions section now highlights the advantages, such as improved supply chain transparency, enhanced traceability, and increased efficiency and cost savings. These benefits are discussed in detail within the paper.
- Addressing Current Challenges and Future Recommendations: We have addressed the current challenges in the field and provided future recommendations within the revised conclusions section. These additions contribute to a more comprehensive understanding of the subject matter and suggest areas for further research and development.
- Impacts of Using BT: We have addressed the impacts of using blockchain technology in the textile and fashion industry. The revised text and conclusions section now explores the transformative potential of blockchain, including its ability to enhance transparency, foster sustainability, and streamline supply chain management.
- Expanded Scope of the Study: We have expanded the scope of the study by providing a more comprehensive methodology and findings sections. These additions offer a broader perspective on the research topic and enhance the overall depth of the study.
- Strengthened Reference List: We have taken your feedback into account and have strengthened the reference list by including more relevant and reputable sources. The current version of the paper includes a total of 98 references.
The changes are marked in blue in the paper, and the revised manuscript is attached to this message.
Once again, we sincerely appreciate your constructive feedback and have made significant revisions to address your comments. We believe these changes have improved the quality and clarity of our article. Thank you for your time and consideration.
Sincerely,
Sunhilde Cuc
Reviewer 2 Report
The paper discusses the very important issue of blockchain technology. Together with the textile industry, it is a very good and new combination.
The paper is written carefully, with all the details necessary. It includes all parts that should be in a good scientific article. The way of the paper is logic and nice to read.
My overall opinion about the reviewed paper is very positive and I think it should be published.
However I have few points that could be improved:
- For Figure 1 I would change the look because it is hard to read the text there.
- Table 1 presents T&T factors - but I do not understand clearly for what reason are these factors? Are they here to share the information or for some other reason? I think it should be written in a more precise way what is the role of these factors.
- I could not see the main goal of the paper. There are many different actions here but no one main aim is highlighted. In my opinion, it would be good to add this information in abstract and introduction.
Author Response
Dear Reviewer,
Thank you for your feedback. I have carefully considered your comments and made revisions to the abstract accordingly. In response to your suggestion, I have updated Figure 2 and included additional text to provide a clearer explanation of the role of traceability and tracking in the context of using blockchain technology in the textile and fashion industry. The changes are marked in blue in the paper, and the revised manuscript is attached to this message.
I appreciate your constructive feedback, and I believe that these changes have improved the clarity of the paper's objectives and methodology. Please do not hesitate to let me know if you have any further suggestions or comments.
Thank you again for your valuable input.
Best regards,
Sunhilde Cuc
Round 2
Reviewer 1 Report
The author have addressed all my comments. I have no further comments.